# Driving through the Lens: Improving Generalization of Learning-based Steering using Simulated Adversarial Examples

## Abstract

To ensure the wide adoption and safety of autonomous driving, vehicles need to be able to drive under various lighting, weather, and visibility conditions in different environments. These external and environmental factors, along with internal factors associated with sensors, can pose significant challenges to perceptual data processing, hence affecting the decision-making of the vehicle. In this work, we address this critical issue by analyzing the sensitivity of the learning algorithm with respect to varying quality in the image input for autonomous driving. Using the results of sensitivity analysis, we further propose an algorithm to improve the overall performance of the task of "learning to steer". The results show that our approach is able to enhance the learning outcomes up to 48%. A comparative study drawn between our approach and other related techniques, such as data augmentation and adversarial training, confirms the effectiveness of our algorithm as a way to improve the robustness and generalization of neural network training for self-driving cars.

## 1 Introduction

Autonomous driving is a complex task that requires many software and hardware components to operate reliably under highly disparate and often unpredictable conditions. While on the road, vehicles are likely to experience day and night, clear and foggy conditions, sunny and rainy days, as well as bright cityscapes and dark tunnels. All these external factors can lead to quality variations in image data, which are then served as inputs to autonomous systems. Adding to the complexity are internal factors of the camera (e.g., those associated with hardware), which can also result in varying-quality images as input to learning algorithms. One can harden machine learning systems to these degradations by simulating them at train time (Chao et al., 2019). However, there currently lacks algorithmic tools for analyzing *the sensitivity of real-world neural network performance on the properties of simulated training images* and, more importantly, *a mechanism to leverage such a sensitivity analysis for improving learning outcomes*. In this work, we quantify the influence of varying-quality images on the task of "learning to steer" and provide a systematic approach to improve the performance of the learning algorithm based on quantitative analysis.

Image degradations can often be simulated at training time by adjusting a combinations of image quality attributes, including blur, noise, distortion, color representations (such as RGB or CMY) hues, saturation, and intensity values (HSV), etc. However, identifying the correct combination of simulated attribute parameters to obtain optimal performance on real data during training is a difficult—if not impossible—task, requiring exploration of a high dimensional parameterized space.

The first goal of this work is to design a systematic method for studying, predicting, and quantifying the impact of an image degradation on system performance after training. We do this by measuring the similarity between real-world datasets and simulated datasets with degradations using the well known Fréchet Inception Distance (FID). We find that the FID between simulated and real datasets is a good predictor of whether training on simulated data will produce good performance in the real world. We also use FID between different simulated datasets as a unified metric to parameterize the severity of various image quality degradations on the same FID-based scale (see Section 3).

Our second goal is to borrow concepts from the adversarial training literature (Madry et al., 2018; Shafahi et al., 2019; Xie et al., 2020) to build a scalable training scheme to improve the robustness of autonomous driving systems against multi-faceted image degradations, while increasing the overall accuracy of the steering task for self-driving cars. Our proposed method builds a dataset of adversarially degraded images by apply evolutionary optimization within the space of possible degredations during training. The method begins by training on a combination of real and simulated/degraded data using arbitrary degradation parameters. On each training iteration, the parameters are updated to generate a new degradation combination so that system performance is (approximately) minimized. The network is then trained on these adversarially simulated images to promote robustness. Our proposed algorithm speeds up the combinatorial degradation updates by discretizing the search space using our FID-based parameterization. See details in Section 4.

Experiments show that the algorithm improves the task performance of "learning to steer" up to 48% in mean accuracy over strong baselines. We compare our approach with other related techniques, such as data augmentation and adversarial training, and the results show that our method consistently achieves higher performance. The method also improves the performance on datasets contaminated with complex combinations of perturbations (up to 33%), and additionally boosts performance on degradations that are not seen during training, like simulated snow, fog, and frost (up to 29%), as discussed in Section 4.2.

For evaluation, we propose a more complete robustness evaluation standard under 4 different scenarios: clean data, single perturbation data, multiple perturbation data, and unseen perturbation data. While previous works usually use one or two scenarios, our work is among the first to test results under all 4 meaningful scenarios that are important for evaluating robustness of algorithms.

We also plan to release "learn to steer under perturbations" datasets for benchmarking. These datasets will contain a base dataset; simulated adversarial datasets with multiple levels of image degradation due to either single or multiple image quality attributes; and simulated adversarial datasets with five levels of combinatorial perturbations due to a different set of unseen factors on images with corruptions in ImageNet-C (Hendrycks & Dietterich, 2019), totaling about 1.2M images and 120 datasets in all. (See Section 4.5.)

## 2 RELATED WORK

The influence of the noise and distortion effects on real images on learning tasks has been explored in the last five years. For example, researchers have examined the impact of optical blur on convolutional neural networks and present a fine-tuning method for recovering lost accuracy using blurred images (Vasiljevic et al., 2016). This fine-tuning method resolved lost accuracy when images were distorted instead of blurred (Zhou et al., 2017). While these fine tuning methods are promising, (Dodge & Karam, 2017b) find that tuning to one type of image quality reduction effect would cause poor generalization to other types of quality reduction effects. Comparison on image classification performance between deep neural networks and humans has been done by (Dodge & Karam, 2017a), and found to be similar with images of good quality. However, Deep Neural Networks struggle significantly more than humans on low-quality, distorted or noisy images. Color spaces have also been shown to negatively affect performance of learning models. It was observed that perturbations affect learning performance more in the Y channel of YCbCr color space that is analogous to the YUV color space (Pestana et al., 2020). Another work (Wu et al., 2020) studies the effect of Instagram filters, which mostly change the coloring of an image, on learning tasks. In this work, we study nine common factors characterizing image quality, i.e., blur, noise, distortion, three-color (Red-Green-Blue or RGB) channels, and hues, saturation, and intensity values (HSV). Not only does our study analyze a more comprehensive set of image attributes that could influence the learning-based steering task, but we also parameterize these nine factors into one integrated image-quality space using the Fréchet Inception Distance as the unifying metric, thus enabling us to conduct sensitivity analysis.

Researchers have also explored how to improve the robustness of learning algorithms under various image quality degradations. One recent work (Tran et al., 2017) provides a novel Bayesian formulation for data augmentation. Cubuk et al. (2018) proposes an approach to automatically search for improved data augmentation policies. Ghosh et al. (2018) performs analyses on the performance of Convolutional Neural Networks on quality degradations due to common causes like compression loss, noise, blur, and contrast, etc. and introduces a method to improve the learning outcomes using

a master-slave architecture. Another work (Hendrycks et al., 2019) shows that self-supervision techniques can be used to improve model robustness that exceeds the performance of fully-supervised methods. A new method, also by Hendrycks et al., tackles model robustness from a data augmentation perspective, where compositions of transformations are used to create new data that is visually and semantically close to the original dataset (Hendrycks et al., 2020). Gao et al. (2020) proposes a technique that re-purposes software testing methods, specifically mutation-based fuzzing, to augment the training data of DNNs, with the objective of enhancing their robustness. A more recent work (Gong et al., 2020) introduces a solution to improving model generalization by first augmenting training dataset with random perturbations, and then minimizing worst-case loss over the augmented data. Our work differs from these studies in several regards. First, we simulate adversarial conditions of image factors instead of commonplace image conditions. Second, we conduct a systematic sensitivity analysis for preparing datasets that are representative of image degradations due to multiple factors at different levels. Third, our algorithm for improving the generalization of the learning task can work with the discretized parameter space while generalize well to the continuous parameter space. Another advantage of our approach is that we can augment the training dataset without the derivatives of the factor parameters, which may not exist or are difficult to compute. These differences distinguish our approach from the prior arts and other techniques for improving model robustness, such as data augmentation and adversarial training.

## 3 ANALYZING IMAGE QUALITY FACTORS

### 3.1 EXPERIMENT SETUP

**Base Dataset**. We build our base dataset using the real-world driving dataset (Chen, 2018), which contains about 63,000 images at the resolution $455 \times 256$. There are several good autonomous driving datasets, but not all of them are suitable for the end-to-end learning to steer task. Some datasets (e.g., Waymo (Sun et al., 2020), KITTI (Geiger et al., 2013), Cityscapes (Cordts et al., 2016), OxfordRoboCar (Maddern et al., 2017), Raincouver (Tung et al., 2017), etc) do not contain steering angle labels. Some other datasets contain steering angle labels (e.g., Audi (Geyer et al., 2020), Honda (Ramanishka et al., 2018), nuScenes (Caesar et al., 2019), Ford AV (Agarwal et al., 2020), Canadian Adverse Driving Conditions (Pitropov et al., 2020), etc), but they are not designed for end-to-end learning to steer task specifically, the driving environment may not be suitable for our end-to-end learning to steer task. For example, driving at intersections (lead to confusion for steering), or only a very small subset of data containing image data from turning on a curved road without branches. There are also several well-known simulators like CARLA (Dosovitskiy et al., 2017) that can generate synthetic dataset, but our work focuses on the real-world driving using real images. Chen's dataset is collected specifically for steering task and has a relatively long continuous driving image sequence on a road without branches and has relatively high turning cases.

**Image quality factors**. We study nine image attributes in this work, i.e., blur, noise, distortion, three-color (Red-Green-Blue or RGB) channels, and hues, saturation, and intensity values (HSV). Blur, noise, and distortion are among the most commonly used and directly affect image quality. R, G, B, H, S, V channels are chosen because both RGB and HSV spaces are frequently used to represent image color spaces: RGB represent three basic color values of an image, while HSV represent three common metrics of an image. Other color spaces such as HSL or YUV have similar properties, hence are excluded from this study to avoid redundant analysis.

**Learning Algorithm**. We choose the model by Bojarski et al. (2016) as the learning algorithm. The model contains five convolutional layers followed by three dense layers. We select this model, because it has been used to steer an AV successfully in both real world (Bojarski et al., 2016) and virtual world (Li et al., 2019).

**Evaluation Metric**. We use mean accuracy (MA) to evaluate our regression task since it can represent the overall performance under different thresholds. We first define the accuracy with respect to a particular threshold $\tau$ as $acc_\tau = count(|v_{predicted} - v_{actual}| < \tau)/n$, where $n$ denotes the number of test cases; $v_{predicted}$ and $v_{actual}$ indicate the predicted and ground-truth value, respectively. Then, MA is computed as $\sum_\tau acc_{\tau \in \mathcal{T}}/|\mathcal{T}|$, where $\mathcal{T} = \{1.5, 3.0, 7.5, 15, 30, 75\}$ contains empirically selected thresholds of steering angles. Based on MA, we use the maximum MA improvement (denoted as MMAI), the average MA improvement (denoted as AMAI), and mean Corruption Errors (mCE) (Hendrycks & Dietterich, 2019) as the evaluation metrics.

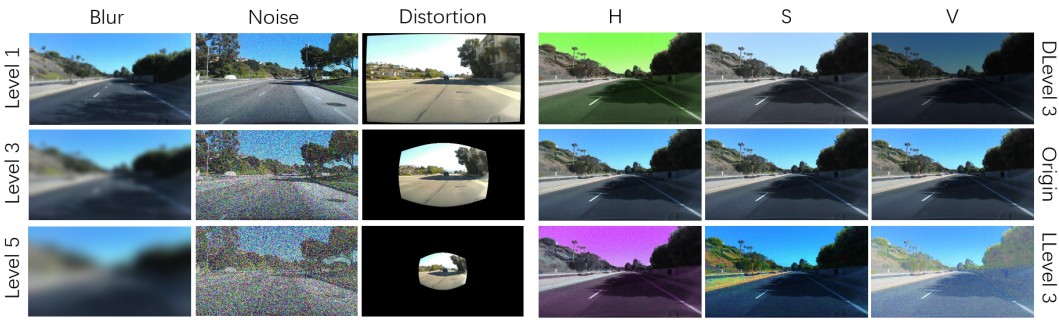

Figure 1: Example images of quality degradation. Five levels of quality reduction are simulated using the blur, noise, and distortion effects (Left) and each lighter and darker direction per channel (Right).

## 3.2 SENSITIVITY ANALYSIS

In this section, we introduce how we conduct sensitivity analysis on the nine image factors, i.e., blur, noise, distortion and R, G, B, H, S, V channels, and how we use the sensitivity analysis results to prepare datasets that are representative of image qualities at different levels for further model training.

For simulating blur, noise, and distortion effects, we use Gaussian blur (Bégin & Ferrie, 2004), additive white Gaussian noise (AWGN), and radial distortion (Zhang, 2000), respectively. For representing channel-level perturbations, we use a linear combination model: denote the value range of one channel as $[a, b]$, in the darker direction, we set $v_{new} = \alpha a + (1 - \alpha)v$; in the lighter direction, we set $v_{new} = \alpha b + (1 - \alpha)v$. The default values are $a_C = 0$ and $b_C = 255$, where $C$ represents one of the 6 channels. To prevent an image being completely dark, we set $a_V = 10$ and $b_H = 179$.

We adopt the Fréchet Inception Distance (FID) as a unified metric for our sensitivity analysis, instead of using the parameter values of each image factor for three reasons. First, FID can better capture different levels of image qualities than the parameter values of each factor, because the correspondence between the factor parameter values and image quality is not straightforward. For example, if we uniformly sample the parameter space of distortion, most parameter values will result in images with qualities similar to the level 4 and level 5 images of distortion datasets shown in Fig. 1. This is not ideal since we would like to have representative datasets that enable us to study the *sensitivity* of a learning task with respect to the gradual degradation of each image attribute. Second, by adopting FID as a single metric, we can map the parameter spaces of all factors into one space that better facilitates both the dependency trend and sensitivity analyses. Lastly, FID can naturally serve as a metric to evaluate the distance between two image datasets. It takes image pixels, image features, and correlations between images into account – these are all meaningful factors to interpret the performance of a learning-based task.

Starting from empirically-selected parameters of each factor, we 'simulate' perturbed datasets and compute their corresponding mean accuracy (MA) using the trained model on the base dataset (Chen, 2018). We then map these correspondences between MAs and the parameter values into the FID space using the FID between each perturbed dataset and the base dataset. Characterizing the influence of each image quality attribute on the task performance, as captured in MA-FID space, we can then re-define parameter value range of each quality factor so that the newly generated perturbed datasets will have similar 'distance gaps' in image quality, as defined by FID. Examples of the resulting images are shown in Fig. 1. More examples can be found in Appendix A.2. The corresponding parameters of the final perturbed datasets can be found in Appendix A.3. The notation for these datasets is summarized as follows: $R$: the base dataset (Chen, 2018); $B1, B2, B3, B4, B5$: five levels of *blur*; $N1, N2, N3, N4, N5$: five levels of *noise*; $D1, D2, D3, D4, D5$: five levels of *distortion*; $RD1, RD2, RD3, RD4, RD5$: five dark levels of the red (R) channel; $RL1, RL2, RL3, RL4, RL5$: five light levels of the red (R) channel; the dark and light levels of the G, B, H, S, V channels have similar naming conventions, i.e., channel label + D/L (dark/light) + level (from 1 to 5).

The analysis results using the final perturbed datasets at different levels of several sample factors are shown in Fig. 2. The complete numeric results and the figure for all factors can be found in Appendix A.6. Note that all factors have some influence on the learning-based steering task. Their

(negative) impact increases, as the perturbation level increases. The performance loss due to image quality degradation can be as high as more than 50% (see lighter Lv5 of the G channel), which can introduce very high risks in autonomous driving.

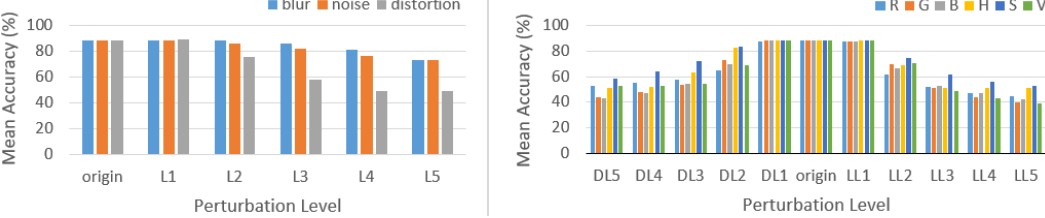

Figure 2: **The relationship between MA and different levels of perturbation:**  Greater image degradations lead to higher loss of mean accuracy.

The final MA differences in the FID space are visualized in Fig. 3 (for blur, noise, distortion, G channel and V channel, see entire figure in Appendix.A.4). Since FID aligns different factors into the same space, we can compare the performance loss due to each factor at various levels. We first observe that the learning-based steering task is more sensitive to the channel-level perturbations (i.e., from R, G, B, H, S, V channels) than the image-level perturbations (i.e., from blur, noise, distortion effects). Second, within the quality factors we analyze, the learning task is least sensitive to blur and noise but most sensitive to the intensity value in V channel. Lastly, for the same color channel, darker and lighter levels appear to have different MA trends, i.e., different accuracy loss at the same FID values can vary.

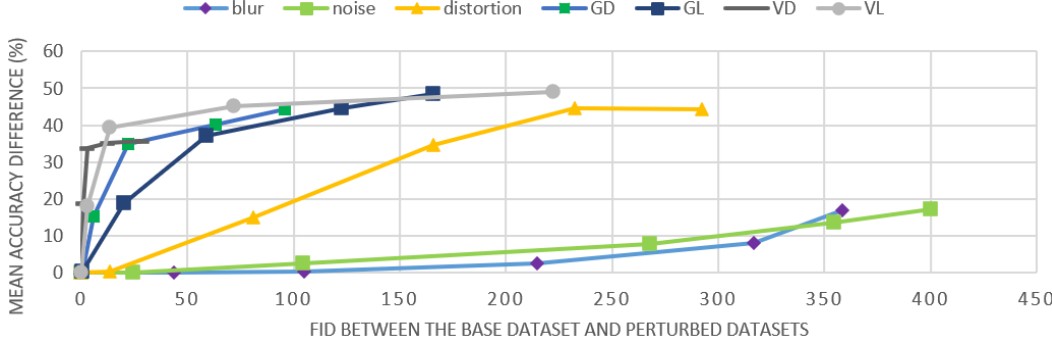

Figure 3: The relationship between FID and MA percentage difference. GD/GL denote G channel in darker/lighter direction, and VD/VL denote V channel in darker/lighter direction.

Compared with other analysis works on learning to steer task, e.g. Tian et al. (2018) and Zhang et al. (2018), our method is the first work that transfers perturbations from multiple parameter spaces into one unified FID space to enable the cross-factor comparison, e.g., to show that our task is more sensitive to V channel perturbation of HSV space than other perturbations.

## 4 IMPROVING GENERALIZATION OF LEARNING-BASED STEERING

### 4.1 METHODOLOGY

In this section, we introduce our method to improve the generalization of the learning-based steering for autonomous driving using the results from the sensitivity analysis.

Our algorithm uses an iterative min-max training process: at each iteration, we first choose a dataset among all datasets of one factor that can minimize mean accuracy (MA). Then, we combine such datasets of all factors with the base dataset to re-train our model while maximizing MA. The algorithm stops when a pre-specified number of iterations is reached or the MA loss is below a certain threshold. The design rationale of our architecture resembles adversarial training: we train the

model to maximize accuracy using the original dataset plus the perturbed datasets with the minimum accuracy in order to improve the robustness of the model. The loss function is the following:

$$\min_{\mathbf{p}} \max_{\theta} MA(\theta, U_{\mathbf{p}}), \tag{1}$$

where $\mathbf{p}$ represents a union of the parameter levels of all analyzed image factors; $\theta$ denotes the model parameters; $U_{\mathbf{p}}$ is the training dataset; and $MA()$ is the function computing MA. The overall description of our method is shown in Algorithm 1. See the overall pipeline figure in Appendix A.1.

Our method has several advantages: 1) the training data is augmented without re-training the model, thus achieving efficiency; 2) it provides the flexibility to generate datasets at various discretized levels of the factor parameters; 3) it does not require the derivatives of factor parameters, unlike other frameworks that optimize factor parameters in the continuous space where derivatives are difficult to compute; and 4) it can generalize over not only the parameters of a single perturbation factor, but also the composition of the parameters of multiple factors.

---

**Algorithm 1:** Improve the generalization of learning-based steering

---

**Result:** a trained model parameterized by $\theta$
Pre-processing:
Conduct sensitivity analysis and discretize the parameters of $n$ factors into their corresponding
 $l_i, i = 1, \ldots, n$ levels;
Generate new datasets for each factor with the discretized values from base dataset $B$:
 $\{D_{i,j}\}, i = 1, 2, .., n, j = 1, 2, ..l_i$;
Initialization:
Set $t = 0$, and initialize the maximum of iterations $T$ and the number of epochs $k$;
Initialize model parameters $\theta^{(0)}$;
**while** $t \leq T$ **do**
  For each factor, select the dataset $D_{i,p_i}$ that can minimize the validation mean accuracy,
    where $p_i = \arg\min_j MA(\theta^{(t)}, D_{ij})$;
  Merge all selected datasets $U_{\mathbf{p}} = (\bigcup_{i=1}^{n} D_{i,p_i}) \bigcup B$;
  Train the network for $k$ epochs and update the parameters $\theta^{(t+1)} = \text{train}(\theta^{(t)}, U_{\mathbf{p}}, k)$ to
    maximize $MA(\theta^{(t+1)}, U_{\mathbf{p}})$;
  Break if stop conditions are met;
  Otherwise set $t = t + 1$;
**end**

---

## 4.2 EXPERIMENTS AND RESULTS

All experiments are conducted using Intel(R) Xeon(TM) W-2123 CPU, Nvidia GTX 1080 GPU, and 32G RAM. We use the Adam optimizer (Kingma & Ba, 2014) with learning rate 0.0001 and batch size 128 for training. The maximum number of epochs is 1000. Images in the base dataset are sampled from videos at 30 frames per second (FPS). For efficiency, we adopt a similar approach as in Bojarski et al. (2016) by further downsampling the dataset to 5 FPS to reduce similarities between adjacent frames. The resulting dataset contains approximately 10,000 images. We randomly split the dataset with about 20:1:2 as training/validation/test data.

We compare our method with three methods: a baseline method (Bojarski et al., 2016), an adversarial training method (Shu et al., 2020), and a naïve data augmentation method. For the naïve data augmentation method, we simply merge all perturbed datasets together to train the model. We test the performance of all methods in four scenarios with increasing complexity:

- Scenario 1: *Clean data*. Test on the base clean dataset only.
- Scenario 2: *Single Perturbation*. Test on datasets of each factor at its discretized levels. Specifically, we use five levels for blur, noise, distortion, and ten levels for R, G, B, H, S, V. Hence, we have 75 datasets in total for testing in this scenario.

- Scenario 3: *Combined Perturbation*. Test on datasets with combinations of all factors at all levels. To be specific, we sample varying levels from each factor, and combine the resulting datasets of all factors into one *combined* dataset. In total, we have six *combined* datasets for testing in this scenario. See examples in the second row of Figure 6, and parameters for each dataset in the appendix section A.3.

- Scenario 4: *Unseen Perturbation*. Test on datasets under previously unseen factors at different levels. The unseen factors are "motion blur", "zoom blur", "pixelate", "jpeg compression", "snow", "frost", "fog" from ImageNet-C (Hendrycks & Dietterich, 2019). We choose these factors because, specifically, "Motion blur" and "zoom blur" can happen during driving; "pixelate" and "jpeg compression" are possible during image processing; and "snow", "frost", "fog" are natural weather conditions that can affect driving experiences. In addition, all these factors can be possibly "simulated" using the nine image factors analyzed in this work. See examples in Figure 4.

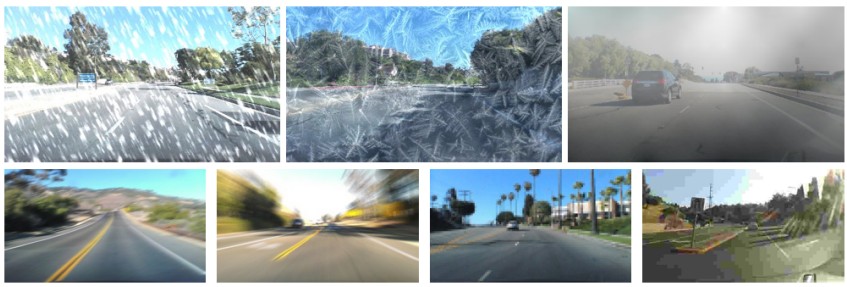

Figure 4: Unseen perturbation examples in our experiments. We use "snow", "frost", "fog" (left to right; first row), and "motion blur", "zoom blur", "pixelate", "jpeg compression" (left to right; second row) from the corruptions in ImageNet-C (Hendrycks & Dietterich, 2019).

We use the maximum MA improvement (denoted as MMAI), the average MA improvement (denoted as AMAI) and mean Corruption Errors (mCE) (Hendrycks & Dietterich, 2019) as the evaluation metrics. From Table 1, we observe that our method outperforms other methods under all metrics in all scenarios. Specifically, our algorithm improves the performance of "learning to steer" up to 48% in MMAI, while reducing mCE down by 50% over the baseline. We have compared our approach with other related techniques, namely data augmentation and adversarial training. The results show that our method achieves consistent better performance. Our method also improves the task performance using the *combined* datasets (Scenario 3) up to 33%. Lastly, when tested on unseen factors (Scenario 4), our algorithm maintains the best performance by 29% in MMAI, while reducing mCE error to 76%.

| | Scenarios | | | | | | | | | |
| --- | --- | --- | --- | --- | --- | --- | --- | --- | --- | --- |
| | Clean | Single Perturbation | | | Combined Perturbation | | | Unseen Perturbation | | |
| Method | AMAI↑ | MMAI↑ | AMAI↑ | mCE↓ | MMAI↑ | AMAI↑ | mCE↓ | MMAI↑ | AMAI↑ | mCE↓ |
| Data Augmentation | -0.44 | 46.88 | 19.97 | 51.34 | **36.1** | 11.97 | 75.84 | 27.5 | 7.92 | 81.51 |
| Adversarial Training | -0.65 | 30.06 | 10.61 | 74.42 | 17.89 | 6.99 | 86.82 | 16.9 | 8.17 | 89.91 |
| Ours | **0.93** | **48.57** | **20.74** | **49.47** | 33.24 | **17.74** | **63.81** | **29.32** | **9.06** | **76.20** |

Table 1: Performance comparison of different methods in four scenarios against the baseline performance, using the maximum MA improvement in percentage (denoted as MMAI), the average MA improvement in percentage (denoted as AMAI), and mean corruption errors in percentage (mCE). For adversarial training, we use the state-of-the-art method described in (Shu et al., 2020). For data augmentation, we simply combine all perturbed datasets into training. As a result, our method outperforms the other methods (highest MA improvements and lowest mCEs) in nearly all scenarios.

We also illustrate the detailed MA improvements in Fig. 5, which shows that our method can achieve improvements in some extreme cases of the channel-factor levels and some of the unseen image effects. However, our method fails to improve on "motion blur", which we plan to study in future. More detailed data, results, and analysis can be found in Appendix A.6.

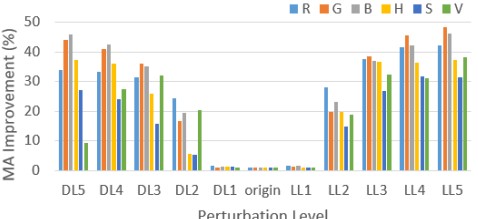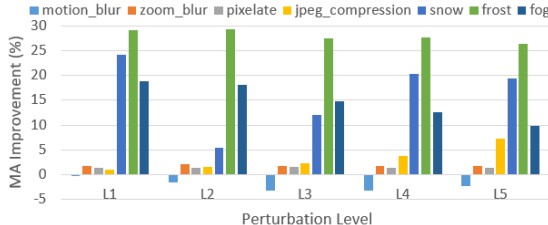

Figure 5: MA improvement with our method compared to the baseline. Our method achieve great improvement on extreme cases for channel-level factors and unseen weather conditions.

By visualizing the salience map on several combined samples in Fig. 6, we show that our method can assist the network to focus on more important areas (e.g., the road in front), instead of focusing on random areas on perturbed images.

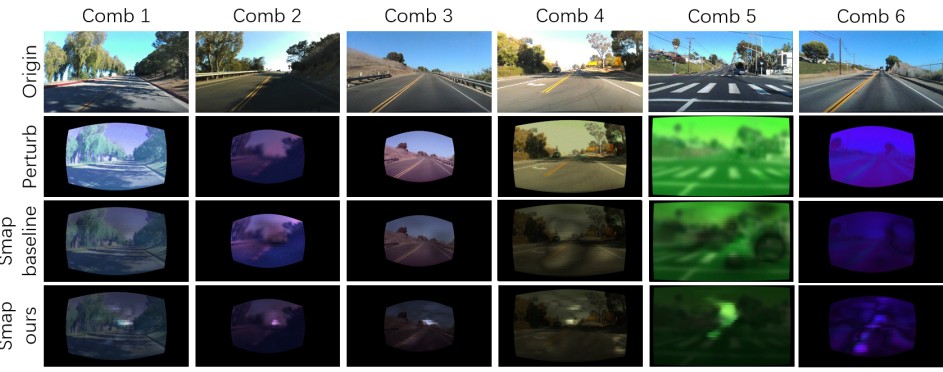

Figure 6: Saliency map samples with baseline method and our method, where the model tests on the different combinations of perturbations. Different columns show different combinations of perturbations. The first row shows the original image, the second row shows the perturbed image with the chosen effects, the third row is the saliency map of baseline model, and the fourth row is the saliency map of our method. It can be seen that, with our method, the network can better focus on the important areas (e.g., the road in front) instead of random areas on the perturbed images, as with the baseline model.

## 4.3 WIDER COMPARISON

To show our method can be generalized well in other cases, we show the experiment results on two new datasets (Honda (Ramanishka et al., 2018) and Audi (Geyer et al., 2020), see dataset setup in Appendix A.5), one new method (Augmix (Hendrycks et al., 2020)), and one new backbone network (comma.ai (Santana & Hotz, 2016)) in Table. 2 and Table. 3. As shown in the tables, our method achieves the best performance on most cases. Specifically, our method outperforms on clean and single perturbation data consistently (at least 2.5% better), and does better in most cases of unseen perturbation data. In combined perturbation data, our method performs better when using NVidia backbone network, while AugMix performs better when using comma.ai backbone network.

## 4.4 GENERALIZATION

In this work, we introduce an efficient and effective computational framework that incorporates sensitivity analysis and a systematic mechanism to improve the performance of a learning algorithm for autonomous driving on both the original dataset and the simulated adversarial scenarios due to multiple perturbations defined on an influential set of important image attributes. This approach can be easily extended and applied beyond the set of factors and the learning algorithm analyzed in this paper. This method can generalize to analyzing any arbitrarily high number of image/input

| | | Scenarios | | | | | | | | |
|---|---|---|---|---|---|---|---|---|---|---|
| | Clean | Single Perturbation | | | Combined Perturbation | | | Unseen Perturbation | | |
| Method | AMAI↑ | MMAI↑ | AMAI↑ | mCE↓ | MMAI↑ | AMAI↑ | mCE↓ | MMAI↑ | AMAI↑ | mCE↓ |
| AugMix+N | -0.12 | 40.64 | 10.94 | 76.48 | 25.97 | 16.79 | 64.41 | **22.23** | 5.99 | 84.95 |
| Ours+N | **2.48** | **43.51** | **13.51** | **67.78** | **28.13** | **17.98** | **61.12** | 16.93 | **6.70** | **80.92** |
| AugMix+C | -5.25 | 55.59 | 9.56 | 86.31 | 31.32 | **0.77** | **106.1** | 37.91 | 7.97 | 89.99 |
| Ours+C | **0.36** | **62.07** | **15.68** | **70.84** | **38.01** | 0.74 | 108.32 | **42.54** | **12.15** | **77.08** |

Table 2: Comparison on Honda dataset. In "Method" column, "N" means Nvidia backbone network, "C" means comma.ai backbone network. Our method outperforms AugMix in most cases.

| | | Scenarios | | | | | | | | |
|---|---|---|---|---|---|---|---|---|---|---|
| | Clean | Single Perturbation | | | Combined Perturbation | | | Unseen Perturbation | | |
| Method | AMAI↑ | MMAI↑ | AMAI↑ | mCE↓ | MMAI↑ | AMAI↑ | mCE↓ | MMAI↑ | AMAI↑ | mCE↓ |
| AugMix+N | -8.24 | 81.89 | 32.22 | 55.27 | 75.49 | 50.23 | 41.98 | 73.06 | 27.39 | 77.51 |
| Ours+N | **4.13** | **94.95** | **45.78** | **18.79** | **80.42** | **59.31** | **29.33** | **75.16** | **31.91** | **42.89** |
| AugMix+C | -18.7 | 57.23 | 12.71 | 89.74 | 36.88 | **28.17** | **68.78** | 37.68 | 4.15 | 119.64 |
| Ours+C | **-1.33** | **77.99** | **30.87** | **50.81** | **47.87** | 23.18 | 73.14 | **42.98** | **14.15** | **80.42** |

Table 3: Comparison on Audi dataset. In "Method" column, "N" means Nvidia backbone network, "C" means comma.ai backbone network. Our method outperforms AugMix in most cases.

factors, other learning algorithms, and multimodal sensor data. Furthermore, other autonomous systems where the perception-to-control functionality plays a key role can possibly benefit from such a technique as well.

### 4.5 BENCHMARKING DATASETS

We plan to release our "learn to steer under perturbations" datasets for benchmarking. The datasets will contain a base dataset, datasets with five levels of perturbation in blur, noise, and distortion, ten levels of variations in the channels R, G, B, H, S, V, multiple with combined perturbations over all nine factors, and five levels of each unseen simulated factor, including snow, fog, frost, motion blur, zoom blur, pixelate, and jpeg compression. There are 120 datasets in total, where each dataset contains about 10,000 images (totaling $\sim$ 1,200,000 images in all). The ground-truth steering angles for all images will also be provided for validation, along with the code that generated all simulated perturbations. We detail the parameters used to generate the datasets in Appendix A.3.

## 5 CONCLUSION AND FUTURE WORK

In this paper, we first analyze how different attributes of image quality influences the performance of the "learning to steer" task for autonomous vehicles, as quantified by mean accuracy (MA). We study three image-level effects (i.e., blur, noise, and distortion) and six channel-level effects (i.e., R, G, B, H, S, V) on real images. We observe that image degradation due to all these effects can impact the task performance. By utilizing FID as the unifying metric, we conduct sensitivity analysis on FID-MA space. Next, we propose an effective and efficient training method to improve the generalization of the model for learning-based steering under multiple perturbations. Our model not only improves the task performance on the original dataset, but also achieves significant performance improvement on datasets with a mixture of perturbations (up to 48%), as well as unseen adversarial examples including snow, fog, and frost (up to 29%). Theses results showing that our model is one of the most general methods in this particular application domain. We will also release the datasets used in this work for benchmarking the evaluation of the robustness of "learning to steer" for autonomous driving.

The implementation of our method currently uses discretization to achieve efficient training, but further implementation optimization is possible. The efficiency can be further improved by utilizing other works like the reweighting strategy(Ren et al., 2018). In addition, we believe that the computational framework is generalizable to other image factors, learning algorithms, multimodal sensor data, and tasks in other application domains. These are all natural directions worthy of further exploration.

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
