# OpenReview forum: "Driving through the Lens: Improving Generalization of Learning-based Steering using Simulated Adversarial Examples"
_ICLR.cc/2021/Conference — Reject_

### Official Review · AnonReviewer4 · 2020-10-26
**Interesting and promising, but the evaluation protocol is not suitable for the task**

**Rating:** 6
**Confidence:** 5

**Review:**

### **Summary**:

This work proposes a new method to improve the generalization of ML models for the task of vehicle steering using a hybrid of data augmentation and adversarial examples. In a nutshell, the proposed method attempts to increase the accuracy of the model by dynamically adding a selection of candidate datasets during training. Each of these “candidates” is created offline applying a transform (e.g. blur, distortion, and changes in color representation) to the original (base) dataset. During training, the method chooses among the K transformed-datasets those who minimize the mean validation accuracy and based on this selection the steering model is retrained. The approach is evaluated on a driving dataset.

---

# Final evaluation (post-rebuttal)


I am increasing the score to "marginally above the acceptance threshold" after the discussion with the authors. I thank the authors for including other datasets and considering additional baselines. I still think that Chen's dataset should not be the main dataset to use in the paper, but I am happy to see strong datasets. I also believe that this paper could benefit from using simulation solutions (despite the domain gap) to create controlled experiments.

---
# Original evaluation (pre-rebuttal)


### **Reasons for score**:

My current score is *rejection*.

I went through the main idea a few times and I think it is a nice proposal. Using the FID as a proxy to cherry-pick transformed datasets seems plausible and relatively simple to reproduce.	After seeing Figure 3. I started having doubts about the actual correlation between MA and FID, which may turn problematic for this approach, but I didn’t let that discourage me too much.

The real concerns emerged when I reached the experiment section. After reading the experiments I think that the narrative of the paper is not as consistent as it could (should?) be.
The paper opens with a clear problem motivation: *“To ensure the wide adoption and safety of autonomous driving, vehicles need to be able to drive under various lighting, weather, and visibility conditions in different environments”*. Also, the title of the paper includes the words *“Improving Generalization”*.

After reading this, I --and probably many readers-- automatically shifted to the mindset of generalization analysis in autonomous driving problems. This typically involves multiple datasets featuring variations such as weather conditions, etc., well-chosen training, validation, and test splits, and strong baselines. However, section 4 does not contain these elements.

All experiments are carried out using Chen’s dataset, and one of the main questions should be, why this dataset? Is this dataset the right one? Honestly, it is hard to tell, because there is not much information about the structure of such dataset. We know that it was collected around Rancho Palos Verdes and San Pedro, (California), but we don’t know much else. How is the dataset split into training, validation, and test? Does it contain strong weather variations? Was the proposed approach tested in a sequence never seen during training or is the method simply overfitting the training routes?

In short, this doesn’t seem to be the right setup t study the problem of generalization. Instead, I would like to recommend a controlled environment, such as the [CARLA simulator](https://github.com/carla-simulator/carla). There you can create datasets in different visual conditions in different maps, and you can also get the steering angle for free. Recent works, such as [Learning Situational Driving](https://openaccess.thecvf.com/content_CVPR_2020/papers/Ohn-Bar_Learning_Situational_Driving_CVPR_2020_paper.pdf) followed this methodology.

---

### **Strengths**:

* This paper is after an interesting and relevant problem, fundamental for the progress of the autonomous driving field.
* Using a possible correlation between MA and FID to drive a dynamic “adversarial data augmentation” process is an interesting idea.
* The paper is well written and simple to reproduce

---

### **Weaknesses**:

* As mentioned above, my main concern is that there are possible flaws in the evaluation protocol. The paper uses Chen’s dataset, but it is not clear if the dataset itself is suitable for the task (besides providing labels for the steering task). I believe that it would be beneficial for the paper to switch to other datasets. Synthetic datasets such as those generated by the CARLA simulator, MS AirSim, or LGSVL simulator should be fine. Real datasets, such as the [AUDI A2D2 dataset](https://www.a2d2.audi/content/dam/a2d2/dataset/a2d2-audi-autonomous-driving-dataset.pdf) could be useful too.

* It seems that strong baselines are not present in the experiments. For instance, I am surprised to see that approaches like [AugMIX](https://arxiv.org/pdf/1912.02781.pdf) or even the original paper by [Hendrycks](https://arxiv.org/pdf/1903.12261.pdf) are not included in the comparison.

---

### **Questions to be discussed during the rebuttal period**:

Please, I would appreciate it if you could refer to the points I described in the **Weaknesses** section.

---

### **Other considerations**:

* The text uses the term master-slave → primary - secondary?

---

---

> ### Author Response · Authors · 2020-11-24
> **Rebuttal for reviewer4**
>
> We sincerely appreciate you for reading our manuscript carefully and providing comments, which helped us improve the paper. While we have addressed all your questions below, we also suggest you refer to the overall key points of our rebuttal above.
>
> About the coverage of the paper: Our work focuses on the perturbations due to natural variations in the environment where input data is captured (the same as “Fuzz Testing based Data Augmentation to Improve Robustness of Deep Neural Networks”). At the same locations, the images will change when some environment factors change, such as lighting and weather conditions.
>
> There are a lot of good autonomous driving datasets, but not all of them are suitable for the learn to steer task. Some datasets (e.g., Waymo, KITTI, Cityscapes, Lyft, OxfordRoboCar, Raincouver, etc) don’t contain steering angle labels. Some other datasets contain steering angle labels (e.g., Audi, Honda, nuScenes, Ford AV, Canadian Adverse Driving Conditions, etc) but part of the driving environment is not suitable for our end-to-end learning to steer task, e.g., driving at intersections (lead to confusion for steering) or only a very small subset of data contains image data from turning on a curved road.
>
> We use Chen’s dataset because it is specifically captured for the image-based steering task (only contains image and steering angle), and has a relatively long continuous driving image sequence on a road without branches and has relatively high turning cases. A shortage of this type of dataset may be low resolution (455, 256) and limit of sensor data (only steering angle), but not the key points of our task. We split the dataset into training, validation and test data, with about 20:1:2.
>
> However, in the new supplementary data, we share and show that our method can be easily verified on other datasets such as Honda and Audi data (2020). Our method can achieve up to over 90% MA improvement on both Honda and Audi, compared to the baseline. The results are consistent with the results on Chen’s dataset.  These results should clearly indicate that our method is generalizable across different datasets -- even when the domains are rather different.
>
> We know there are some great simulators for autonomous driving like CARLA, DeepDrive, Udacity. But we didn’t use pure simulated images because of the domain gap between the 3D graphically rendered synthetic images and real-world images.
>
> In the original paper for ImageNet-C Hendrycks compared networks for classification, while here we have a regression task. Consequently, we only use the ImageNet-C code and their setting to generate the corruptions, but didn’t use their data and experimental results.
>
> AUGMIX is an interesting work. The difference to our work is that AUGMIX generates augmented data without getting information from the target network training process, while our method can adjust the augmentation policy dynamically according to the current training status (validation results) of the target network. We compared our method with AUGMIX on two new datasets (Honda and Audi), and one new backbone network (comma.ai) in Table.2 and Table.3 in Sec.4.3 in the revised paper. As shown in the tables, our method achieves the best performance in most cases, and the improvement can go up to 90% in certain extreme cases.
>
> As a further comparison, Sensei can also adjust the augmentation policy dynamically, but it is an in-training data augmentation approach, while our data augmentation procedure is off-training, which can save the augmentation time during training while still lead to great improvement in robustness. Our work is able to do the data augmentation offline because of two reasons: 1) we show that training on limited discretized augmentation can still improve the test performance when adding random perturbations drawn from a continuous space; 2) we use sensitivity analysis to minimize the discretized level of each perturbation, which minimize the size of generated datasets and make our solution scalable. One issue with the discrete method is the size of the generated dataset, which can be improved by resorting to an online augmentation approach.
>
> Last but not least, we proposed a more complete robustness evaluation standard under 4 different scenarios. Some of previous work focused on the test results under our Scenario 1 (i.e., clean dataset), or Scenario 2 (each test image only has one seen perturbation), or Scenario 4  (each test image only has one unseen perturbation), while our work is the first to propose the test results under all 4 meaningful test scenarios (the above-mentioned 3 scenarios and a new scenario where each test image has multiple combined perturbations). We believe all of them are important for testing robustness of algorithms and should be considered in testing.

---

> > ### Comment · AnonReviewer4 · 2020-11-24
> > **Regarding Chen's dataset**
> >
> > I am interested in knowing if Chen's dataset adheres to  ICLR ethics standards. Is data anonymized? e.g. people faces, license plates, etc?

---

> > > ### Author Response · Authors · 2020-11-24
> > > **Data anonymization**
> > >
> > > Thank you for raising an excellent point.
> > >
> > > Chen's dataset is not anonymized. But since it only provides low-resolution images (455x256), it's not easy to see people's faces and license plates. We can anonymize the data, as needed, before we release our datasets.
> > >
> > > Moreover, the additional datasets (Audi 2020 and Honda 2018) cited and tested in our revised paper are both anonymized. And our method also achieved equally good performance on these two datasets.

---

> > > > ### Author Response · Authors · 2020-11-25
> > > > **ICLR ethics standards**
> > > >
> > > > We certainly agree that privacy in dataset usage is important.  However we are not aware of any ethical rules at ICLR that require the posted dataset to be anonymized, as we review the posted ICLR ethics standards:  https://iclr.cc/public/CodeOfEthics. A number of important and widely used datasets (e.g. ImageNet) are also not anonymized.
> > > >
> > > > As stated above, Chen's dataset has not been explicitly anonymized, however the low-resolution (455x256) of the dataset combined with the wise-angle view makes it extremely difficult if not impossible to discern identities and license plates due to the high level of pixelation.  We will be happy to ensure that published data/images in the paper are not identifiable.  Thank you again for bringing the privacy issue to our attention.

---

### Official Review · AnonReviewer3 · 2020-10-27
**Review for Paper2677**

**Rating:** 4
**Confidence:** 3

**Review:**

This paper proposed a novel adaptive data augmentation algorithm that produces random perturbations on the training dataset to train an imitation learning-based self-driving network. It starts with a sensitivity analysis of network performance under different types and levels of perturbations. And a novel automated perturbed training dataset selection mechanism is then proposed to improve the performance. Validation has been conducted over simulated data with both seen and unseen perturbation types.


Pros:
* Very practical task. Such problems do exist in real-world self-driving, and having learnable perturbations could potentially reduce the generalization gap.
* Great efforts to push and simulate perturbations that could be seen in the real-world.

My primary concern is that there are quite a few very relevant problem settings tackling similar tasks. And the paper should add discussions and comparisons. These include but are not limited to:
- Learning to reweight different training samples. e.g. [A]
- Automated data augmentation with perturbations, e.g. [B]
- Bayes opt for hyper-params search overall, e.g. [C, D]

"Unseen" perturbations are also re-assembling of several low-level degradation factors. Thus it's not clear whether such a domain gap is significant or not. It will help to understand the question if there exists similar perturbation in training. This could be done by choosing the most similar perturbation type in training to each unseen perturbation in testing in terms of image similarity score between the same images under two different perturbations.

The baselines used in the experiment are a bit too weak. Please consider adding the ones as mentioned above. Please consider the "oracle" with a training set that reproduces the perturbations used in testing. This is to evaluate whether this proposed method could improve performance when there is no domain gap in perturbation. For instance, could you even perform better on rainy days through a model trained on the proposed training set rather than a dataset of rainy day logs?

The method was not evaluated on a real dataset. Many real datasets, such as OxfordRoboCar, Raincouver, Canadian Adverse Driving Conditions, contain real-world driving data with challenging weather conditions. Plus, it would be preferable to evaluate perception performance as well, such as detection/segmentation, not only an end-to-end IL driving model, which is less practical at this moment for real-world self-driving.

The incremental dataset augmentation is greedily selected based on the worst validation performance on that single subset. As a consequence, an over-difficult subset might be preferred. Instead, the subset improves overall performance. Could you comment on that?

Minors:
* Getting the full validation performance at each step could potentially time-consuming. Have you considered improving the efficiency by early stopping criteria?
* How many time steps do you run in practice?
* Does the # of training iterations and controlled to be the same across all competing methods to ensure fairness?
* How about the additional run-time compared against no dataset selection?

[A] Ren et al. Learning to Reweight Examples for Robust Deep Learning
[B] Cubuk et al. Autoaugment: Learning augmentation policies from data
[C] Tran et al. A Bayesian Data Augmentation Approach for Learning Deep Models
[D] Snoek et al. Practical Bayesian Optimization of Machine Learning Algorithms



-------------------------------------------------------- Post Rebuttal --------------------------------------------------------------------------------------------

I carefully read the reviewer's comments and the rebuttal. I think the author did not get my major concern on competing algorithms. I acknowledge this proposed method is a novel problem setting and my goal is not to ask for summarizing the difference in problem settings. However, the technical approach, which is based on (batch-)training sample selection and reweighting, could be validated through comparing against more comprehensive baselines, as I pointed out, under similar settings.

Furthermore, I am not convinced by the claim that unseen perturbation cannot be justified if there exists a similar perturbation assemble during training. The reference-based perceptual similarity is a well-studied domain which could be directly adopted to evaluate the overall "difficulty" or "out-of-distribution"-level. And I am not convinced by the author that the experiments you have tried cannot be shown in paper due to page limit: at least you could include in supplementary, as they are important to justify if your experiment setting really validates generalization ability.

In sum, I will keep my current rating.

---

> ### Author Response · Authors · 2020-11-24
> **Rebuttal for reviewer3**
>
> We sincerely appreciate you for reading our manuscript carefully and providing comments, which helped us improve the paper. While we have addressed all your questions below, we also suggest you to refer to the overall key points of our rebuttal above.
>
> We contrast our work against the set of missing references mentioned below:
>
> (A) proposes a novel meta-learning algorithm to assign weights to training examples based on their gradient directions, which can help reduce overfit to training set biases and label noises. By comparison, our method can help improve the robustness of the task performance on different types of perturbations.
>
> (B) proposes an approach to automatically search for improved data augmentation policies, which needs to train a child network for validation and a policy controller for policy generation first, then augment and train on the target dataset.
> In contrast to this work (B), our method selects the augmentation policy and trains the target network at the same time, which can reduce the computation time. Also, in B, the policy is fixed during the training of the target network, while ours can be adjusted dynamically according to the validation result.
>
> (C) provides a novel Bayesian formulation for data augmentation. The key contribution is  extending a GAN model with one data generation model and two discriminative models. However, it cannot include known perturbations that are related to the driving task like various weather conditions. By including the known influential factors, we aim to improve the robustness of autonomous driving.
>
> (D) proposes a practical Bayesian optimization method to reduce the hardness of tuning learning parameters and model hyperparameters. In contrast, ours aims to improve the robustness of the driving task. We believe that the correlation of (D) with our work is relatively weak.
>
> It’s hard to define the similarity of two perturbations. The image similarity of two perturbed datasets can be represented by the pixel-level difference, but whether these two datasets are similar to the neural network is unknown. To the best of our knowledge, the image-level difference does not necessarily lead to the same trend of performance difference.
>
> Our work shows training on certain low-level factors can improve the performance on certain unseen factors. But handling unseen factors is not a small topic that can be clarified by one or two simple experiments, so we didn’t include this part because of the page limitation. Actually we have performed cross validation experiments for training on those different perturbed datasets separately and tested them separately, which is the “oracle” as you mentioned. But we found some out-of-expectation results showing that certain factors have quite different properties. We didn’t include this part because it’s not a simple, consistent conclusion. We plan to document these expansive, long experimental results in an independent technical report.
>
> For dataset selection, see point 2 in the overall official comment. We also conducted experiments and performed more comparison on two new datasets, one concurrent work, and a new backbone, see point 1.
>
> Also, in our earlier investigation, we in fact conducted some segmentation experiments, but we excluded them to not distract the readers from a more focused topic.
>
> Yes, we have observed that an over-difficult subset might be preferred and fixed this issue. Instead of selecting the worst case from each factor, we also tried selecting the worst cases from all factors. By doing this, some very challenging factors are preferred, e.g., Lv5 distortion, making the network incapable of performing well on other factors. However, if we select the worst case from each factor, we no longer have this problem. We observe that within one factor, if we train on the clean data and high level perturbation together, the performance on low level perturbations can also be increased. Lastly, after training several rounds on high-level perturbations, the network can perform better on the high-level perturbation than the lower ones. After that, the algorithm will choose lower level perturbations to train.
>
>
> Other Minor issues:
>
> We use a small validation set, and we only do validation once per iteration (k epochs). Yes early stopping criteria can be used to improve the validation efficiency.
>
> The training time takes about two days using NVIDIA GTX 1080.
>
> Yes, the number of training epochs are controlled. We train all mentioned methods from scratch instead of copying their performance from existing studies.
>
> It depends on the number of factors n. For one epoch, it will be n times longer compared to the baseline method. However, since most of the perturbed datasets contain the key image information for steering, our method uses far less epochs than the baseline method. Overall, our method (n=9) takes about 2~3x longer than the baseline method.

---

### Official Review · AnonReviewer2 · 2020-10-28
**An interesting subminssion that can be improved further**

**Rating:** 4
**Confidence:** 4

**Review:**

Summary:
This paper presents an algorithm to improve the model generalization of the task of "learning to steer". First, the sensitivity of a baseline learning algorithm to degraded images in varying qualities caused by different factors is carried out. Some empirical insights are gained. Then, a new training algorithm is proposed to solve a min-max optimization problem, where the most difficult datasets are chosen and used for training at each iteration. Experiments are conducted to validate the effectiveness of the proposed method.

Pros:
1) The idea of evaluating the sensitivity of a model to different degradation factors in the same metric space is interesting, providing empirical insights for preparing datasets with different degradation levels.
2) Inspired by adversarial learning, choosing the most difficult datasets for training at each iteration may be informative and potentially improve the model generalization.
3) Empirical study on a base dataset for the "learning to steer" task is carried out.

Cons:
1) The choice of FID as the unified metric for evaluating the quality of different degraded images caused by different factors is not convincing. Although the authors list some reasons on page 4, it is still unclear how FID could be a fair metric to different factors. Referring to Figure 3, the mean accuracy difference is no so sensitive to blur and noise according to the FID metric. Is it possible that FID is more sensitive to such degradation factors than others? A careful inspection and more discussions are encouraged. Besides, it is recommended to carry out a user study to register the FID or other metrics to the human visual experience.

2) It seems that the sensitivity analysis is only used as an empirical guideline to discretize the parameters of degradation factors into their corresponding levels. However, the effectiveness of such a guideline has not been validated. The authors can compare it with other discretization methods such as uniformly sampling, unevenly sampling towards heavy degradation, or light degradation. Moreover, if the abovementioned user study is carried out, how is the performance using the discretization method based on the human visual experience levels? Besides, will it be helpful to determine the dataset volumes of different factors, i.e., adjusting the ratio (importance) between different factors? And will the ratio have a significant impact on the performance?

3) The proposed method seems to be not limited to the "learning to steer" task. Considering that some well-established benchmark datasets such as ImageNet-C and different methods have been proposed for studying the impact of degradation factors, it is recommended to carry out experiments on these datasets and included more representative state-of-the-art methods into the comparison.

4) It is unclear how the training cost is since each dataset should be evaluated separately at each iteration in the proposed algorithm. Detailed analysis and comparison are encouraged.

5) On page 5, it is said that the original dataset is also used for training together with the perturbed datasets. However, in the algorithm, only U_p is used, which is defined as the combination of the selected perturbed datasets. Is it the same as the one in Eq. (1)? Besides, what is the meaning of U in the algorithm?

6) In the Experiments Part, both the baseline method and Scenario 1 are called the baseline, which may be confusing in the following description and discussions.

7) Why the model generates strong responses at the distant areas rather than the near road, e.g., markers and boundaries, in the first three columns of Figure 6? Does the vanishing point serve as a cheap feature for "steering"? If so, the structural information of the road should be more useful. Thereby, some degradation factors such as channel perturbations may have a smaller impact on the performance than others such as blur and distortion. However, from Table 2 and Table 3 in the appendix, it seems that distortion and channel perturbations have a larger impact on the generalization performance compared with blur and noise, e.g., at the higher levels. Moreover, the proposed method can effectively deal with the degradation factors that have a smaller impact on structural information such as noise and channel perturbations (i.e., a larger improvement) while being less effective for those structural-related ones (i.e., blur and distortion). Should the degradation factors be divided into different groups, i.e., structural-related ones and structural-irrelevant ones? Will the proposed method have a consistent superiority on the two groups over the common data augmentation technique? More experiments and analysis should be carried out to provide further evidence on the choice of the baseline steering model and benchmark dataset, as well as evaluate the effectiveness of the proposed method comprehensively.

---

> ### Author Response · Authors · 2020-11-24
> **Rebuttal for reviewer2**
>
> We sincerely appreciate you for reading our manuscript carefully and providing comments, which helped us improve the paper. While we have addressed all your questions below, we also refer to the overall key points of our rebuttal above.
>
> 1. In order to explore the sensitivity of the driving task to various image factors, we need a metric to evaluate the difference of two image datasets. We use FID in this work. Other metrics can also be used. More experiments by using the L2 distance (L2D), which we believe is better associated with the human visual experience as suggested, is added to the revised paper. Nevertheless, FID can differentiate various image effects better than L2D, as shown in Appendix A.4 of the revised paper (the curve difference in FID figure is visually more apparent than that in L2D figure), since it can also capture feature information in DNN beyond the pixel-level information.
>
> 2. Using the sensitivity analysis, we can cover the target parameter range of an image factor with the *minimal* number of levels. Sensitivity analysis informs us where the critical cutoff should be (similar with unevenly sampling towards heavy degradation), thereby minimizing the number of perturbation levels per factor. Some other methods such as uniform sampling may need more discretizing levels to achieve the same performance, e.g., we tried on uniform sampling with twice as many levels -- this necessarily leads to twice as many datasets to train, thus more training time.
>
>   Here our sensitivity analysis is performed on each factor to minimize the discretizing level number. If the ratio between different factors needs to be adjusted, an across-factor sensitivity analysis would be required. We believe the ratio will influence the performance.
>
> 3. Our algorithm is not limited to the “learning to steer” task, but the focus of this work is on autonomous driving. In addition to the algorithm, our approach includes many task-specific designs for the driving task, such as the choice of training and test factors, the evaluation metric (not just MSE for general regression task, instead FID), etc. They are all valuable components for the task “learning to steer”. Actually we also did some segmentation experiments, but we excluded them due to the page limit and topic focus.  However, to show our method can be generalized well in other cases, we show the new experiment results on two new datasets (Honda and Audi), one new method (AugMix), and one new backbone network (comma.ai) in Table.2 and Table.3  in Sec.4.3 of the revised paper. As shown in the tables, our method achieves the best performance in most cases, and the improvement can go up to 90% in certain extreme cases.
>
> 4. Overall the training process takes about 2 days on a GTX1080 GPU. We evaluate our approach using a small validation dataset, which costs much less than the k-epoch training time in 1 iteration. We will add detailed descriptions.
>
> 5. The original dataset should be included in training with the perturbed dataset. This is a typo in Algorithm 1, and we have fixed this typo in the revised paper.
>
> 6. Scenario 1 is the clean base dataset. We will clarify this confusion.
>
> 7. It’s difficult to explain why the neural network focuses on the vanishing point of the road, but, compared to the baseline method’s random attention, it’s an improvement.
>
>   The structural information of the road should be more useful according to our experience, since we know the color of the road will not influence the road structure. But for the network, it’s possible that it uses color to find the vanishing point, e.g., just the top point of the road color area.
>
>   Yes, it’s an excellent idea to explore how the structure information influences performance.  In fact, we’ve already done extensive exploration along this line, but to keep this paper focused on “image perturbations” within the 8-pages limit of ICLR, we excluded this part of our investigation.

---

### Official Review · AnonReviewer1 · 2020-11-02
**Adversarial example generations of self-driving car**

**Rating:** 4
**Confidence:** 5

**Review:**

The paper aims to generate adversarial examples for self-driving car. It has two fold contributions. First, the paper analyzes the sensitivity of a learning algorithm w.r.t. different image transformation in a realistic scenario. Next, they also propose a technique to learn from the generated images.

The paper addresses a very important problem. The reported performance also quite good. However, there are some major concern about this work:

1.	There are many work along this line in other communities. For example,
a.	Tian, Yuchi, et al. "Deeptest: Automated testing of deep-neural-network-driven autonomous cars." Proceedings of the 40th international conference on software engineering. 2018.
b.	Zhang, Mengshi, et al. "DeepRoad: GAN-based metamorphic testing and input validation framework for autonomous driving systems." 2018 33rd IEEE/ACM International Conference on Automated Software Engineering (ASE). IEEE, 2018.
The above two papers work on the same premise that the self-driving car learning model is susceptible to many real-world conditions and propose methods to analyze to the sensitivity of the method. Especially, the second paper uses GAN to simulate image close to the original distributions. The paper should compare with these baselines.

2.	 The re-training method using evolutionary optimization is interesting. However, a similar technique is proposed by Gao, Xiang, et al. "Fuzz testing based data augmentation to improve robustness of deep neural networks." Proceedings of the ACM/IEEE 42nd International Conference on Software Engineering. 2020. The author should compare the retraining method with this technique.

3.	Figure 4 generated images that do not look very real to me.

4.	Only tested for one learning model.

---

> ### Author Response · Authors · 2020-11-24
> **Rebuttal for reviewer1**
>
> We sincerely appreciate you for reading our manuscript carefully and providing comments, which helped us improve the paper. While we have addressed all your questions below, we also suggest you read the overall key points of our rebuttal above.
>
> 1. Deeptest proposes a systematic testing tool for detecting erroneous behaviors of DNN-driven vehicles that can potentially lead to fatal crashes. Although it also uses real-world factors to generate perturbed images and detects erroneous images that affect the driving tasks, it does not focus on analysing the sensitivity of the driving task to these factors.
>
>   DeepRoad proposes a GAN-based approach to generate driving scenes under various weather conditions and tests inconsistent behaviors of the autonomous driving system. However, they do not analyze the influence of different levels of weather conditions on the driving task.
>
>  Both studies focus on erroneous behaviors detection. Our work differs from them by proposing a systematic method to study, predict, and quantify the impact of image degradations on the performance of “learning to steer”, e.g., our work can inform the sensitivity of the network performance in steering due to perturbations in various factors affecting the quality of input images. Our method is the first work that transfers perturbations from individual parameter spaces into one unified FID space to allow the cross-factor comparison.
>
> 2. Sensei is an in-training data augmentation approach, while our data augmentation is off-training. Our approach can save the computation time of the augmentation process during training while still lead to substantial improvements in robustness. Our work is able to perform data augmentation offline for two reasons: 1) we show training on limited discretized augmentation can still improve the test performance when adding random perturbations drawn from a continuous space; 2) we use sensitivity analysis to minimize the number of discretized levels of perturbation for each factor, which minimizes the size of generated datasets and makes our solution scalable. One limitation of our method is the size of the generated dataset, which can be improved by resorting to an online augmentation approach. This feature highlights the flexibility of our approach between disk space and computation time.
>
> 3. The content of the images are all real and we use the same setting and code of ImageNet-C to generate image effects.
>
> 4. The model we chose is a representative model in the task of end-to-end driving. Many other studies of end-to-end driving have adopted a very similar network architecture. So, we only test one model in this work.  We also believe that this idea is generalizable as we show by testing on newer datasets that appeared this year.
>
>   We show the new experiment results on two new datasets (Honda and Audi), one new method (AugMix), and one new backbone network (comma.ai) in Table.2 and Table.3 in Sec.4.3 of the revised paper. As shown in the tables, our method achieves the best performance in most cases, and the improvement can go up to 90% in certain extreme cases.

---

### Review · Ethics_Committee · 2020-12-31

**Decision:**

Concerns raised (can publish with adjustment)

**Ethics Review:**

The paper makes use of Sully Chen's Driving dataset https://github.com/sullychen/driving-datasets , which is a dataset of continuous driving image sequences recorded from a car dashcam with labeled steering angles.  Images depict the street, looking out from the car onto the road.

The authors apply perturbations and plan to release their “learn to steer under perturbations” datasets for benchmarking.

Because of the use of this dataset, R4 and the Area Chair note that the submission may conflict with ICLR Ethics standards, and in particular flag concern with the "respect privacy" principle.

The authors point out that the ethical considerations with respect to this dataset may be comparable to the ethical considerations in other widely used datasets, such as ImageNet, that are also not anonymized.

Factors to consider include:
1. Licensing and copyright of the original dataset
2. Licensing and copyright of the datasets released with this paper
3. ICLR Ethics guidelines for "Uphold High Standards of Scientific Excellence", including consent from Human Subjects
4. ICLR Ethics guidelines for "Privacy", including anonymization

----
Breakdown:

1. Licensing and Copyright issues of the original dataset seem minimal: This data has been released under an MIT License, permitting the stated uses of the dataset here.

2. Licensing and Copyright of the new dataset is not specified.

3. Scientific Excellence: Releasing the versions of the dataset created in this paper follow scientific excellence goals of reproducibility and fair comparisons in benchmarking.

4. Privacy: I cannot find images where a person, or a person's car or home address, appears to be uniquely identifiable. This would be the most serious concern for privacy. That said, it may be the case that there exists some image in this dataset where some person may be able to identify someone who does not wish to be identified, e.g., based on knowledge of a depicted person's basic characteristics, the location where the image was taken, etc. This would be an aggregate approximate biometric based on multiple cues. The foreseeable harms from this include a bad actor finding someone who would like to be hidden.  The authors state that they will take privacy into account when they release the dataset specifically with respect to these concerns: They volunteer to blur faces/licenses where they are identifiable.

----
Recommendations:

The dataset in question, as far as I can tell, was constructed from public spaces, meaning that sharing the images publicly is not inappropriate outright. The questions of concern here mostly have to do with the potential unique identification of individuals against their will or without their consent.  However, this is not a dataset of people, unique identifiers are not annotated, and the dataset was not constructed to make unique identification possible. Additional checks/edits on the dataset to handle uniquely identifiable content would be reasonable and permit everything to proceed with minimal negative effects.

In more detail:

The supplementary material does not include the dataset that the authors say they will release, nor the license for that dataset.  It is therefore not possible to make a fully informed judgement on the main ethical values at play with respect to this dataset.  Ideally, details on the ethicality would not proceed unless the dataset were there to actually be examined.

Assuming the authors can indeed blur everything that might identify an individual, and create an appropriate license, then there are minimal foreseeable risks.  If the authors do not blur content, but have a reasonable license, the foreseeable harm is still relatively low and unlikely compared to the foreseeable benefit of advancing work in this area in a rigorous, reproducible way.

Ideally, the license should stipulate that it does not permit sharing uniquely identifying personal content without the explicit permission from the identifiable person.  Thus, any images shared publicly -- e.g., in papers, talks -- would be limited to those that are not uniquely identifying.

Even more ideally would be more rigorous methods for handling dataset distribution and removal of data instances in benchmark datasets, but that is beyond the scope of this work.

---

### Author Response · Authors · 2020-11-24
**Overall key points of our rebuttal**

We sincerely appreciate the reviewers for reading our manuscript carefully and providing comments, which helped us improve the paper. We have addressed all questions and comments, and revised the paper accordingly (including comparison & other recently published concurrent works). At a high level, we would like to highlight 4 key issues in the rebuttal: additional results, dataset, method, and test scenarios.

First, to show our method can indeed be generalized well in other cases, we show the new experiment results on two new datasets (Honda and Audi) as suggested by reviewers, one new method (AugMix), and one new backbone network (comma.ai) in Table.2 and Table.3  in Sec.4.3 of the revised paper. As shown in the tables, our method achieves the best performance in most cases, and the improvement can go up to 90% in certain extreme cases.

Second, explanation for dataset selection. There are several good autonomous driving datasets, but not all of them are suitable for the learning-to-steer task. Some datasets (e.g., Waymo, KITTI, Cityscapes, Lyft, OxfordRoboCar, Raincouver, etc) do not contain steering angle labels. Some other datasets contain steering angle labels (e.g., Audi, Honda, nuScenes, Ford AV, Canadian Adverse Driving Conditions, etc) but part of the driving environment is not suitable for our end-to-end learning-to-steer task, e.g., driving at intersections (lead to confusion for steering) or only a very small subset of data contains image data from turning on a curved road. We use Chen’s dataset because it is specifically captured for the image-based steering task (only contains image and steering angle), and has a relatively long continuous driving image sequence on a road without branches and has relatively high turning cases. A weakness of this type of dataset may be its relatively low resolution (455, 256), and limited types of labels (only steering angle), although neither of these things interfere with our investigation -- in fact, they provide some of the worst-case scenarios for real-world applications.

Third, comparison with some other recent work. AugMix generates augmented data without getting information from the target network training process, while our method can adjust the augmentation policy dynamically according to the current training status (validation results) of the target network. Sensei can also adjust the augmentation policy dynamically, but it is an in-training data augmentation approach, while our data augmentation procedure is off-training, which can save augmentation time during training while still lead to great improvement in robustness. Our work is able to do the data augmentation offline because of two reasons: 1) we show that training on limited discretized augmentations can still improve the test performance when adding random perturbations drawn from a continuous space; 2) we use sensitivity analysis to minimize the number of discretization levels for perturbation in each factor, which minimizes the size of generated datasets and makes our solution more scalable. One issue with the discrete method is the size of the generated dataset, which can be improved by resorting to an online augmentation approach. We added most of the references mentioned by reviewers as references to analyze relationships with our methods or for comparison in the revised paper.

Last but not least, we proposed a more complete robustness evaluation standard under 4 different scenarios. Some of previous work focused on the test results under our Scenario 1 (clean dataset), or Scenario 2 (each test image only has one seen perturbation), or Scenario 4  (each test image only has one unseen perturbation), while our work is the first to propose the test results under all 4 meaningful test scenarios (the above-mentioned 3 scenarios and a new scenario where each test image has multiple combined perturbations). We believe all of them are important for testing robustness of algorithms and should be considered in testing.

Please find our detailed response for each reviewer below.

---

### Decision · Program_Chairs · 2021-01-07
**Final Decision**

**Decision:**

Reject

**Comment:**

This meta-review is written after considering the reviews, the authors’ responses, the discussion, and the paper itself.

The paper has 2 main contributions: 1) analysis of the sensitivity of a deep network predicting steering angle from images w.r.t. different synthetic image perturbations, 2) A training method, based on adaptively adjusted data augmentation, which improves the robustness of a model to seen and previously unseen perturbations.

The reviewers’ opinions are somewhat mixed, leaning towards negative. The reviewers point out that the task is important and the methodology makes sense, but the experiments are limited: only one dataset, only synthetic perturbations, only the steering angle prediction task (which is not necessarily very practical), not strong enough baselines, no results on established datasets like ImageNet-C. The authors addressed some of these issues in the updated version of the paper (more datasets, one more baseline), but most reviewers did not change their evaluation.

Based on all this information and reading the paper itself, I recommend rejection at this point. The paper has interesting ideas, but the experimental evaluation is not sufficient. Moreover, I find the use of the steering prediction task confusing - there does not seem to be anything driving-specific in the method, so using standard datasets (like ImageNet-C) would be more convincing. For driving datasets, using real-world (not synthetic) image perturbations would be advisable. As the paper stands, it looks neither like a proper application paper, nor as a fundamental method/analysis paper, but something inbetween, which is not to its favor.